# Multimodal Physical Exercise and Functional Rehabilitation Program in Oncological Patients with Cancer-Related Fatigue—A Randomized Clinical Trial

**DOI:** 10.3390/ijerph20064938

**Published:** 2023-03-10

**Authors:** Eduardo J. Fernandez-Rodriguez, Celia Sanchez-Gomez, Roberto Mendez-Sanchez, Jose I. Recio-Rodriguez, A. Silvia Puente-Gonzalez, Jesus Gonzalez-Sanchez, Juan J. Cruz-Hernandez, Maria I. Rihuete-Galve

**Affiliations:** 1Department of Nursing and Physiotherapy, University of Salamanca, 37007 Salamanca, Spain; 2Instituto de Investigación Biomédica de Salamanca (IBSAL), 37007 Salamanca, Spain; celiasng@usal.es (C.S.-G.);; 3Department of Developmental and Educational Psychology, University of Salamanca, 37005 Salamanca, Spain; 4Department of Medicine, University of Salamanca, 37007 Salamanca, Spain; 5Medical Oncology Unit, University Hospital of Salamanca, 37007 Salamanca, Spain

**Keywords:** cancer, quality of life, exercise, multidisciplinary, rehabilitation, disability

## Abstract

The increase in life expectancy and survival time implies an increase in the possible side-effects of pharmacological treatments in patients. Cancer-related fatigue is one of these side-effects. The main objective of this study was to evaluate the effects of a multimodal program of physical exercise and functional rehabilitation on asthenia, pain, functional capacity, and quality of life in cancer patients with cancer-related fatigue. Methods: This was a randomized, parallel-controlled clinical trial, with two arms (experimental and control group), and it was conducted over the course of a year in the Oncology Hospitalization Unit at the University Hospital of Salamanca, Spain. Participants (n = 48) were assessed at three points during the study. The first assessment was prior to hospital discharge, the second assessment was after 15 days, and the final assessment was at one month post-hospital follow-up. The intervention lasted one month. The main variables studied were the dependency levels (Barthel), cancer-related fatigue (FACT-An), health-related quality of life (EuroQoL-5D), functional capacity (SPPB), and kinesiophobia (TSK-F). Results: Sample size (n = 44). Mean age 63.46 ± 12.36 years. Significant differences between control and experimental group participants in Barthel, FACT-An, TSK-F, and SPPB scores at follow-up and final assessment. Conclusions: There are beneficial effects of a multimodal physical exercise and functional rehabilitation program in improving the autonomy of cancer-related fatigue patients.

## 1. Introduction

Advances in the treatment and care of cancer patients have led to an increase in five-year survival in cancer. The increase in life expectancy and survival time implies an increase in people living with side-effects of pharmacological treatments [1].

Cancer-related fatigue, pain, dyspnea, and depression are the main symptoms secondary to cancer treatment [2]. These symptoms result in decreased functionality and quality of life associated with increased survival in cancer patients [3].

Among these symptoms, cancer- and treatment-related asthenia is the most frequent, and it very negatively affects the quality of life of patients [4]. Cancer-related fatigue is not considered with the importance it deserves by some diagnostic and therapeutic approaches in oncology because it is considered to be a normal symptom [5,6].

Although asthenia is sometimes confused with fatigue, it is important to clarify that fatigue is only a symptom of asthenia [7]. Fatigue is defined as tiredness or exhaustion as a result of physical or mental exertion; however, asthenia is tiredness or exhaustion in the absence of physical or mental exertion [8].

Cancer-related fatigue, or the feeling of asthenia, limits and affects the cancer patient’s participation in daily life and activity. This asthenia is usually associated with other alterations, such as cachexia, sarcopenia, and anorexia, due to their pathophysiology links [5], and with some body composition parameters, such as loss of muscle mass or increase in visceral fat [9].

Some pharmacological treatments have shown beneficial effects on cancer-related asthenia, although with the presence of some adverse side-effects [6,10,11]. Therefore, it is important to investigate the implementation of oncologic treatment and care with non-pharmacologic interventions, which have also shown beneficial effects. This therapeutic approach is supported by the expert panel of the National Comprehensive Cancer Network (NCCN), which considers educational measures, controlled therapeutic physical exercise, and energy conservation techniques as fundamental within a comprehensive functional rehabilitation program [12]. In addition, evidence from recent studies supports the use of physical exercise, physiotherapy, and occupational therapies, which seem to improve asthenia [6,10,11,13,14].

A complete functional rehabilitation program should be multimodal and have an interdisciplinary biopsychosocial approach (oncologist, nursing, physiotherapy, and occupational therapy) [15,16]. For this reason, it is advisable to use different tasks and to associate physical exercise with cognitive work and other psychosocial components [15,16]. In relation to psychosocial factors, it is known that chronic pain or fatigue syndrome favor the presence of fear-avoidance behaviors related to movement and/or physical activity, which provoke or exacerbate these symptoms [17,18,19].

Because of the clinical complexity of the advanced cancer patient, the selection of the optimal intervention environment is fundamental in their treatment, to try to favor the patient’s ability to self-manage their situation. In previous studies, supervised physical exercise interventions outside the health care setting, both in the community and at home, have shown good results [19,20].

However, there is currently a lack of studies of multimodal physical exercise and functional rehabilitation programs to assess their effects in patients with cancer-related fatigue outside health care settings. In a health care setting, we can mention the study by Kröz et al. on the influence of a multimodal and multimodal-aerobic therapy concept on health-related quality of life in breast cancer survivors [21].

This study aimed to increase knowledge about the possible effects of an interdisciplinary multimodal physical exercise intervention and a home-based functional rehabilitation program used to develop effective models of care that promote the recovery of personal autonomy in cancer patients with asthenia after hospital discharge, in order to improve their quality of life.

The main objective of this study was to evaluate the effects of a multimodal program of physical exercise and functional rehabilitation on asthenia, pain, functional capacity, and quality of life in cancer patients with cancer-related fatigue.

## 2. Materials and Methods

### 2.1. Design and Aims

This study was a randomized, parallel-controlled clinical trial, with experimental and control groups, and it was conducted in the Oncology Hospitalization Unit at the University Hospital Complex of Salamanca (CAUSA). The protocol of the clinical trial was previously approved by the Clinical Research Ethics Committee of the Area of Health of Salamanca (ID: PI 202007547) and the trial was conducted in accordance with the Declaration of Helsinki. The clinical trial was registered in ClinicalTrials.gov (INCT04761289). (18 February 2021).

The aim of this study was to assess the effects of a new clinical standard practice treatment with a multimodal physical exercise and functional rehabilitation program on asthenia, pain, functional capacity, and quality of life in patients with cancer-related fatigue versus isolated application of the standard clinical practice treatment.

For recruitment, consecutive sampling was used to select patients with oncological asthenia who were hospitalized in the Oncology Unit at that time. The recruitment period was from July 2021 to February 2022.

### 2.2. Participants

Participants were recruited at the Oncology Hospitalization Unit at the time of hospitalization, and were included in the study after voluntarily signing the informed consent and meeting the selection criteria. Inclusion criteria: Diagnosis of an oncological disease, over 18 years of age, hospitalized in the Oncology Hospitalization Unit, score between 15 and 55 points on the Barthel Index (BI), score of 4 or more on the visual analog scale (VAS) for cancer-related asthenia, and voluntarily signed the informed consent for participation in the study. Exclusion criteria: A cognitive status lower than 23 points on the Mini Mental State Examination (MMSE), which makes it difficult to understand the indications during the study, and a hemoglobin level lower than 10 g/dL. Withdrawal criteria: Progression of the oncologic disease leading to an end-of-life stage of illness or death, and failure to complete the follow-up and final evaluations.

Randomization, allocation, and blinding: The recruited subjects, once they met the selection criteria, were randomly assigned to one of the two study groups (Figure 1). Randomization was performed by an investigator external to the evaluations using a simple randomization method with a table of random numbers generated using Microsoft Excel 2021.

Participants were not blinded due to the nature of the intervention in each group. However, to increase the rigor of the study, the investigators who performed the assessments and those who conducted the statistical analysis were blinded and were unaware of the assignment of the participants to the two groups.

Sample size: The sample size was estimated using the program EPIDAT 4.2, accepting an alpha risk of 0.05 and a beta risk of 0.2 in a two-sided test. Twenty-five subjects were necessary in first group and twenty-five in the second to recognize as statistically significant a difference greater than or equal to 7.5 units in the Barthel Index score, which was considered as a primary variable. The common standard deviation was assumed to be 8.3 units [22,23] and a drop-out rate of 20% was anticipated.

### 2.3. Procedures and Data Collection

Each participant performed an initial visit after recruitment and before randomization, and two follow-up visits, one fifteen days after baseline, considered as the intermediate evaluation, and the final evaluation at one month after baseline assessment.

### 2.4. Primary and Secondary Outcomes

We considered the Barthel Index score as the primary variable to assess the degree of dependence in activities of daily living (ADLs). The following secondary variables were considered as secondary variables in the study: cancer-related asthenia, health-related quality of life (HRQoL), pain, functional capacity, fear/avoidance of movement as a psychosocial factor associated with asthenia and pain.

### 2.5. Variables and Measurement Instruments

Barthel Index (BI) [24]: It is used to assess the degree of dependence in activities of daily living (ADLs). It has demonstrated its validity and reliability, and it is a questionnaire that is easy to apply and interpret. Score from 0 to 100, middle range 15–55 points.

Fourth version of the Functional Evaluation of Cancer Therapy-Anemia (FACT-An) [25]: It is used to assess cancer-related fatigue and functional capacity. It is considered to be the most appropriate for the assessment of the symptom. The FACT-An scale consists of the FACT-G fatigue subscale plus seven non-fatigue-related items. In this study we used the FACT-G version for cancer-related fatigue. Score from 0 to 188.

EuroQoL 5-D Questionnaire (EQ-5D) [26,27]: This questionnaire is used to assess the health-related quality of life (HRQoL). It has been adapted and validated for use in the Spanish population. Score from 5 to 15; in the “euro-QoL thermometer, 0 to 100”.

A visual analogue scale (VAS) [28,29]: This scale is the most widely used to subjectively assess pain. The intensity of pain is represented on a 10 cm line, where 0 indicates no pain and 10 represents the worst possible pain. Values lower than 4 mean mild or mild-moderate pain; from 4 to 6 indicate moderate–severe pain; and more than 6 indicates very intense pain.

Short Physical Performance Battery (SPPB) [30]: This test is used to assess physical performance. It includes three parts that assess balance, gait speed, and lower limb strength. It is used to classify the degree of frailty in the elderly and to predict adverse events, dependency, institutionalization, and even mortality. Score from 0 to 12.

The Tampa Scale of Kinesiophobia–Fatigue (TSK-F) [18,31]: This questionnaire was developed to assess fear of movement related to fatigue and pain. This is a symptom that is often underestimated and is little studied in cancer patients. Its efficacy has been demonstrated in oncological patients and patients with chronic fatigue syndrome. We used the 11-item version (TSK-F-11). Score from 11 to 44.

All measurements taken were recorded on a data collection sheet for each patient and then recorded in a database designed for this study.

Interventions: The intervention was carried out by an interdisciplinary team specialized in oncology patients. The team consisted of physiotherapists, occupational therapists, nurses, and oncology doctors. Participants in both study groups received pharmacological treatment and specific care, as well as guidelines adapted to their specific health needs. Prior to hospital discharge, all participants received a health education program and materials promoting an active and healthy lifestyle as part of the intervention. This program consisted of an informative talk and a guideline with written indications.

The experimental group, in addition to the conventional care, participated in an intervention implemented with a month-long multimodal program of physical exercise and functional rehabilitation at home. This program consisted of:

1. Multimodal physical exercise program:

A supervised and structured multimodal physical exercise program was carried out at home for one month. Two short sessions of 15–20 min per day were performed, one in the morning and one in the afternoon. The sessions followed the 2010 American College of Sports Medicine (ACSM) recommendations for the structure of each session [32,33] (initial warm-up (2–3 min), multicomponent physical exercise (8–12 min), and cool-down and relaxation (5 min)). We also followed the recommendations of the 2019 ACSM Guidelines for Exercise and Cancer update [16], adapting some on FITT (frequency, intensity, time, type). We increased the frequency of weekly sessions but with a shorter duration because of a lower tolerance to exercise and effort due to the characteristics of our study sample, i.e., patients recently discharged with cancer-related fatigue.

The sessions were mainly based on a combination of aerobic and resistance training. The warm-up consisted of aerobic exercises designed to promote joint mobility and muscle activation. The cool-down consisted of aerobic training with mild muscle stretching and relaxation exercises. The main phase of the program combined aerobic and balance exercises with moderate-load resistance exercises. These were targeted to upper and lower quadrant muscle groups with 2 sets of 4–8 repetitions. In the exercise program, mainly in the resistance and balance exercises, the dose and load were individually adapted according to the initial and intermediate assessment and, followed by a progression in difficulty and load, were coordinated in a global workout encompassing the major muscle groups at least 3–4 times during the week.

In the exercise program, mainly in the resistance and balance exercises, the dose and load were individually adapted according to the initial and intermediate assessment and followed a progression in difficulty and load, maintaining a moderate intensity in their performance according to a perception of moderate exertion on the Borg scale (score 4–6/10). The program was supervised by an external researcher contracted for this task in a face-to-face clinical evaluation.

2. Functional rehabilitation program:

This program consisted of re-education of activities of daily living. Before hospital discharge, specific training was carried out to identify factors that hindered or prevented the performance of ADLs: (1) direct intervention on ADLs performed in the hospital that was generalized to the patient’s daily environment; (2) simplifying activities and teaching energy-saving techniques (EST); and (3) sleep hygiene counseling according to specific NCCN guidelines on cancer-related fatigue [34]. In the specific program, work was done with patients to improve their Basic Activities of Daily Living (BADLs), considering them as essential to lead as normal a life as possible despite the clinical and functional situation of the individuals.

3. Environmental adaptations and the need for assistive products: In addition, prior to discharge, we assessed whether assistive products could be prescribed to support personal autonomy and identified potential barriers in the home. This was achieved by visiting the patient’s own home and observing first-hand the difficulties that restrict normal daily activities.

Work plan and visits: As shown in the flowchart in Figure 1, participants in both groups were required to complete 3 assessment visits: 1 at baseline, an intermediate visit at 15 days, and a final visit at 1 month. All assessments were planned in the same way, with each visit lasting approximately one hour. All outcome variables were measured at all three study visits and they were conducted by the same investigators, who did not know to which group each subject was assigned in the study.

Baseline visit: The initial evaluation was performed before patient discharge, checking the inclusion and exclusion criteria at baseline. At this visit, sociodemographic data, medical history, comorbidities, and pharmacological treatments of the patients were recorded, and as in the rest of the visits, all the outcome variables of the study were collected. Randomization and assignment to the study groups was performed once the baseline visit was completed, and the necessary indications were given to the patients in each of the groups.

Intermediate and final visits: At these visits, all the outcome variables collected at the baseline visit were recorded again. The follow-up visits, at 15 days and 1 month after the start of the study, were carried out in the Clinical Teaching Assistance Unit of the Faculty of Psychology at the University of Salamanca.

Data analysis: The study data were collected in a database using a unique identification code for each study participant. This database did not contain personal or contact information for any of the participants. Baseline characteristics of the study population were expressed as mean ± standard deviation (SD) for quantitative variables and as frequency distribution for categorical variables.

Baseline data were analyzed to calculate differences between groups using an independent-samples t-test for quantitative variables, and the chi-squared test for qualitative variables.

The statistical analysis was carried out on an intention-to-treat basis with two-way repeated measures ANOVA. Unadjusted and baseline-adjusted models were used in cases in which the baseline value had a statistically significant difference in the baseline assessment between groups (FACT, FACT other concerns, SPPB, SPPB gait speed). The presence or absence of sphericity was considered and Greenhouse–Geisser correction was performed when necessary. Hypothesis testing established an α of 0.05. Data were analyzed with the SPSS Statistics version 26.0 software (IBM Corp, Armonk, NY, USA).

## 3. Results

All prospectively selected patients who met the study inclusion criteria were included in the study. The study had a final sample of 48 individuals: 24 individuals in the experimental group and 24 individuals in the control group. The mean length of hospital stay was 8 days. The time between initial and final assessment was 30 days. The demographic distribution of the individuals in the study is shown in Figure 1.

Table 1 shows the sociodemographic characteristics of the patients at the start of the study. We observed that age, weight, height, body mass index, number of treatment lines, and Charlson comorbidity index were very similar in the groups under study. In both groups, the male gender predominated over the female gender and primary education predominated over secondary and higher education. The sample (n = 48) was composed of 30 men (14 experimental group; 16 control group) and 18 women (10 experimental group; 8 control group). The mean age of the total sample was 63.46 years (±12.45), and was similar in both groups. The body mass index (BMI) in the experimental group was 22.58 (±3.23), with an average weight of 65.70 (±13.41) and an average height of 169.92 (±7.76), whereas the BMI in the control group was 22.75 (±3.56), with an average weight of 65.61(±13.45) and an average height of 168.83 (±8.61). Regarding the Charlson comorbidity index, we observed similar scores in both groups. The experimental group presented an average score of 9.50 (±4.12), while the control group had an average score of 9.96 (±3.18).

In terms of anatomopathological diagnosis, the predominant type of cancer was lung cancer (52%), followed by tumors of the digestive system (23%), prostate cancer (12%), and breast cancer (11%) (other tumor types 2%). Furthermore, in terms of tumor stage, 64% of the sample was at stage III, 23% at stage IV, and 13% at stage II.

Table 2 shows the descriptive results of the total score obtained in the variables under study. Similar scores were observed in both groups at baseline assessment.

Analyzing the most significant global scores in the study, we observed a mean Barthel Index in the experimental group of 47.08 (±10.47), and in the control group of 40.63 (±12.79), showing a moderate level of dependence of the patients.

Regarding the FACT-An questionnaire, individuals in the experimental group had a mean score of 94.96 (±12.79), while in the control group it was 102.75 (±13.00).

Analyzing the quality of life, the EuroQoL-5D questionnaire showed a score of 10.33 (±1.78) in the experimental group, while in the control group the mean score was 10.88 (±2.19).

We also observed a mean SPPB score of 6.21 (±2.99) for individuals in the experimental group, and a mean score of 4.42 (±3.03) for individuals in the control group.

Finally, regarding the analysis of the TAMPA kinesiophobia scale scores, individuals had a mean score of 20.25 (±7.07) in the experimental group and 18.7 (±5.90) in the control group.

In addition, the rest of the more specific scores, as mentioned above, can be seen in Table 2.

### Primary and Secondary Outcomes

Table 3 shows the mean scores and the comparison of means at the three points in time for the different variables. In addition, the degree of statistical significance is included to establish whether the changes during follow-up are statistically significant, and report a considerable improvement in the patients in the intervention group.

Analyzing the different variables under study, we can observe the following:

For levels of dependency, we observed statistically significant differences in the Barthel Index scores obtained. These differences were evident in the experimental group in the follow-up (*p* < 0.001) and final assessment (*p* < 0.001), while in the control group, statistically significant differences were observed in the follow up (*p* = 0.015), but not in the final assessment (*p* = 0.180), where a worsening in the dependency levels of the participants was observed, as can be clearly seen in Figure 2.

For levels of cancer-related asthenia, we observed statistically significant differences in the scores obtained on the FACT scale, both in the total score and in the specific scores (physical condition, social environment, emotional state, personal functioning, and other concerns). These differences were evident in the experimental group in the follow-up (*p* < 0.001) and final assessment (*p* < 0.001), while in the control group, a worsening of the participants’ cancer-related asthenia levels was observed, as can be clearly seen in Figure 2.

With regard to the results concerning the parameters of health-related quality of life, we observed a statistically significant improvement (*p* < 0.001) in the patients in the control group at follow-up (15 days) and final assessment (30 days), while, on the contrary, in the experimental group, we observed a regression in the levels of quality of life measured with the EuroQoL-5D questionnaire (*p* > 0.05) at both times of the study. On the contrary, observed that the “EuroQoL thermometer” shows statistically significant improvements (*p* < 0.05) in the participants of the experimental group compared to those of the control group. Analyzing this questionnaire more closely in terms of its specific items, we were unable to identify statistically significant differences in the items “personal care”, “daily activities”, “pain”, or “anxiety”. These results are shown in Figure 3.

Regarding the VAS data, we observed similar results for VAS pain and VAS fatigue. We found statistically significant differences, consistent with clinical improvement, in the follow-up and final assessment in the experimental group (*p* < 0.001), while in the control group, we saw improvements in the follow-up assessment (*p* < 0.05), but a statistical worsening in the final assessment (*p* > 0.05). These results are shown in Figure 3.

Functional capacity: Analyzing the results of the SPPB test, we observed statistically significant differences in the total score in the two assessments (follow-up and final) in the experimental group (*p* < 0.001), whereas in the control group, no such differences were observed. This was also observed in the different specific items of the scale, with the exception of the item “gait”, where statistically significant differences were observed in the experimental group (*p* < 0.001) and in the control group (*p* < 0.05). These results are shown in Figure 2.

Finally, in terms of the analysis between groups, considering differences in the follow-up and in final evaluation, we observed:−For the level of dependence, we found statistically significant differences at the time of follow-up (*p* = 0.034; *p* < 0.05) and at the time of the final assessment (*p* < 0.001), between the experimental and control groups.−For cancer-related asthenia levels, we found statistically significant differences at follow-up (*p* = 0.019; *p* < 0.05) and at the time of the final assessment (*p* < 0.001), between the experimental and control groups.−Regarding the levels of health-related quality of life, we found statistically significant differences at follow-up (*p* = 0.096; *p* < 0.05) and at the time of the final assessment (p < 0.001), between the experimental and control groups.−For pain levels, measured with the VAS, we observed that there were no statistically significant differences between the two groups under study, neither at follow-up (*p* = 0.552; *p* < 0.05) nor at the final assessment (*p* = 0.398; *p* < 0.05).−In terms of physical performance, we found statistically significant differences at the follow-up (*p* = 0.009; *p* < 0.05) and at the final assessment (*p* < 0.001), between the experimental and control groups.−Finally, analyzing the levels of kinesiophobia, we observed statistically non-significant differences at the time of follow-up (*p* = 0.754; *p* < 0.05), but, on the contrary, we observed statistically significant differences at the time of the final evaluation (*p* = 0.031; *p* < 0.05), between the experimental group and the control.

## 4. Discussion

Principal findings:

The study showed that the implementation of a one-month post-hospital follow-up with a controlled physical exercise and functional rehabilitation program in oncology patients with cancer-related fatigue achieved functional improvement (Barthel) and decreased cancer-related fatigue (FACT) in patients. However, not all individuals showed a direct relationship between these improvements and health-related quality of life (EuroQoL-5D).

In recent years, and due to the exponential increase in survival of cancer patients [35], we have observed a consequent increase in interest in the symptomatic control of these patients, since as a result of the different treatments or the progression of the tumor process, side-effects appear more frequently, with cancer-related fatigue being the most incidental and prevalent according to the scientific literature [4].

The multimodal exercise program was adapted in terms of the frequency and duration of the exercise sessions with respect to the recommendations of the latest update of the ACSM (2019) [16]. This modification does not conform to the evidence proposed by other authors [36], who state that fatigue reductions are greater with exercise sessions longer than 30 min and programs of more than 12 weeks compared to less exercises. However, we consider that this modification is justified by the characteristics of our sample, i.e., patients recently discharged from the hospitalization period with moderate cancer-related fatigue. Patients with cancer-related fatigue present a lower exercise tolerance [37], so we decided to perform more weekly sessions by reducing the time of aerobic exercises and sets/repetitions of resistance exercises. Our study supports the need to always consider the patient’s characteristics and previous situation, as a starting point, in order to apply the general recommendations for the performance of therapeutic exercise. Moreover, as the experts report in the ACSM guideline 2019 update [16], the reviewed studies did not always enroll individuals with low values in outcomes, as occurred in our patients with moderate to high cancer-related fatigue. Therefore, sticking to the FITT prescription may or may not generalize to cancer survivors in greatest need.

Analyzing in more detail the variables related to the levels of dependency of the individuals, we observed a clear benefit of the implementation of the specific post-hospital follow-up program for oncology patients. All individuals showed statistically significant improvements after the implementation of the proposed functional rehabilitation program. This is clearly corroborated by the results of similar studies in other incapacitating pathologies such as COPD or dyspnea [23].

We consider it essential that clinical improvement is generalized with the correct performance of activities of daily living for cancer patients once they are discharged from the hospital complex. We believe that this improvement in terms of autonomy not only benefits the patient themselves, but also the family environment surrounding them, freeing them from assuming the role of caregiver. In fact, we are considering this as a line of future research. Furthermore, as a novelty in this study, we consider it appropriate to prescribe individualized support products, such as walkers or standing frames, which have shown a greater benefit in terms of the autonomy of cancer patients than in recent studies in the scientific literature [38,39,40].

With regard to the levels of cancer-related fatigue, we observed that the implementation of the post-hospital follow-up program achieved improvements in these levels, substantially reducing the levels of cancer-related fatigue. Furthermore, a direct relationship was established between this symptomatic improvement and an improvement in terms of dependency. An analysis of the existing literature shows that this correlates directly with studies by Campbell, Covington, and Sleight and his collaborators [16,41,42].

A comprehensive analysis of the results related to health-related quality of life reveals some contradictions. In the total scores of the questionnaire, we observed statistically significant improvements in the control group, whereas, in the scores of the general item, “EuroQoL thermometer”, we found that these differences were manifested in the experimental group. In the existing literature, we found similar results to those presented in the “EuroQoL thermometer” [43,44], but we raise the possibility that the measuring instrument, a generic instrument for the assessment of people with any pathology, may have caused biases in the measurement. Therefore, in future studies, we propose the use of specific instruments for measuring quality of life in oncology patients in order to be able to draw better conclusions about this variable under study.

Continuing with the analysis of the variables under study, we found clinical improvements in the functional capacity of the individuals, and in levels of kinesiophobia, which, together with a reduction in the intensity of pain in individuals in the experimental group, leads us to believe that these three parameters may be directly related. One possible hypothesis related to these results is that a reduction in pain will lead to less kinesiophobia, which will have a direct impact on the improvement in the functional capacity of oncology patients. Pergolotti and colleagues, in their CARE study, highlighted the importance of the implementation of a functional rehabilitation program with controlled physical exercise in this type of patient, and how this improved their functional capacity, with positive repercussions for their dependency parameters [43]. We should highlight the low scores obtained with the SPPB questionnaire, which we believe may be related to the age of the individuals and their level of comorbidities.

Furthermore, in a systematic review by McTiernan and colleagues, the most relevant conclusion was that the implementation of physical activity programs in oncology patients leads to a lower risk of side-effects and improved survival [44].

In our study, we proposed three levels of intervention depending on the degree of symptomatic affectation presented by the patient when prescribing the different exercises in the rehabilitation program, with the possibility, obviously, of making modifications depending on the evolution of the individual (increasing or decreasing the degree of complexity). We believe that greater precision in the prescription of the rehabilitation program can bring us even closer to better clinical practice, as mentioned by Pergolotti and his collaborators in the EXCEEDS study, in which they proposed an algorithm, “Exercise in Cancer Evaluation and Decision Support (EXCEEDS)” [45]. They developed this to improve clinical decision making in oncological rehabilitation and to facilitate access to these multimodal physical exercise programs in cancer patients. They used the Delphi methodology to assess usability and acceptability and determine pragmatic priorities for the implementation of clinical interventions. This is also in line with other major studies in the existing scientific literature, as mentioned by Kroz et al. [46].

The current results show the benefit of a rehabilitative intervention in the follow-up of post-hospital patients, and how this type of intervention leads to improvements in terms of dependency, functional capacity, and decreased levels of cancer-related fatigue in cancer patients. These results are clinically relevant for several reasons. Firstly, as it is a clinical intervention outside of the hospital environment, and can even be conducted in the individual’s own home, this process of humanization of health care activity is enhanced. Secondly, and taking into account this symptomatic improvement of patients, the intervention provides not only a direct benefit for them; in addition, it also provides indirect benefits, both for their closest relatives, as no extra care will be necessary, and for health policies, preventing possible acute exacerbations of the symptoms treated that lead to a clinical worsening resulting in an earlier hospital admission, with the consequent increase in the need for health resources.

Adherence to the prescribed intervention was complete in all individuals in the sample. These data provide greater strength to the study, and we believe adherence is conditioned by the pathology of the individuals, which perhaps contributes to a greater degree of acceptance of the users to the different treatments.

In future studies, we consider it essential to carry out this clinical practice of post-hospital follow-up in other types of symptoms related to the oncological disease, such as dyspnea or pain, or in other words, to proceed with a generalization process, which would allow all cancer patients to benefit from this proposed post-hospital functional rehabilitation program.

It is important to note the main limitations of this study. Due to the nature of the intervention, it was not possible to completely blind the participants; however, a recent meta-epidemiological study suggested that blinding is less important than is commonly believed [47]. On the other hand, a greater equality of the sample in the initial parameters would allow us to obtain even better conclusions. The duration of the intervention was only 1 month, so the long-term sustainability and effect of the intervention could not be measured. Furthermore, although subjects were advised at baseline and follow-up visits that they could not receive other types of external rehabilitative interventions, we cannot guarantee that they would not be used. Finally, a last possible limitation is conditioned by adherence to the prescribed intervention. All the patients stated that they performed the exercises as prescribed, but this is something we cannot be sure of, as it is only their confidence in what they tell us that allows us to corroborate this.

Finally, we consider it important to highlight the important potential for generalization of the results obtained in the study. We believe that these clinical practices should be included in the routine intervention of this type of oncology patients.

## 5. Conclusions

The results of this study support the beneficial effects of a multimodal physical exercise and functional rehabilitation program carried out by an interdisciplinary team specialized in the clinical intervention of oncology patients, made up of oncologists, nurses, occupational therapists, and physiotherapists, in improving asthenia levels, functional capacity, pain, and quality of life in the autonomy of cancer-related fatigue patients.

## Figures and Tables

**Figure 1 ijerph-20-04938-f001:**
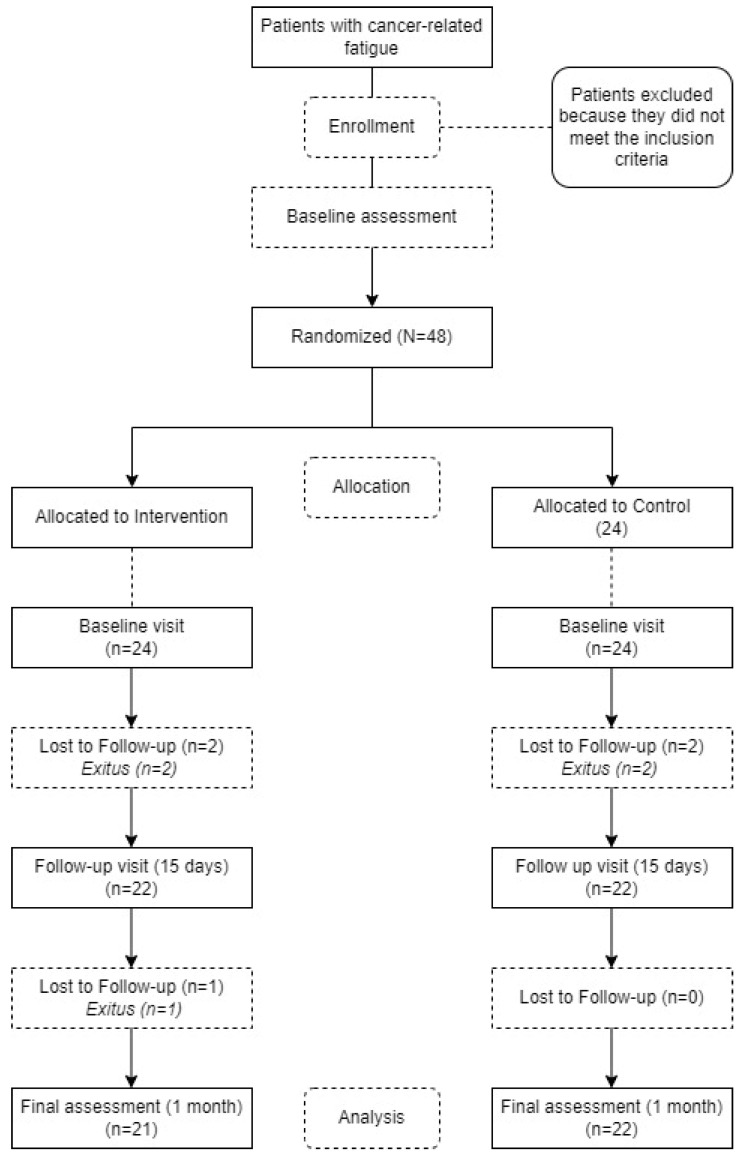
Flowchart of the study.

**Figure 2 ijerph-20-04938-f002:**
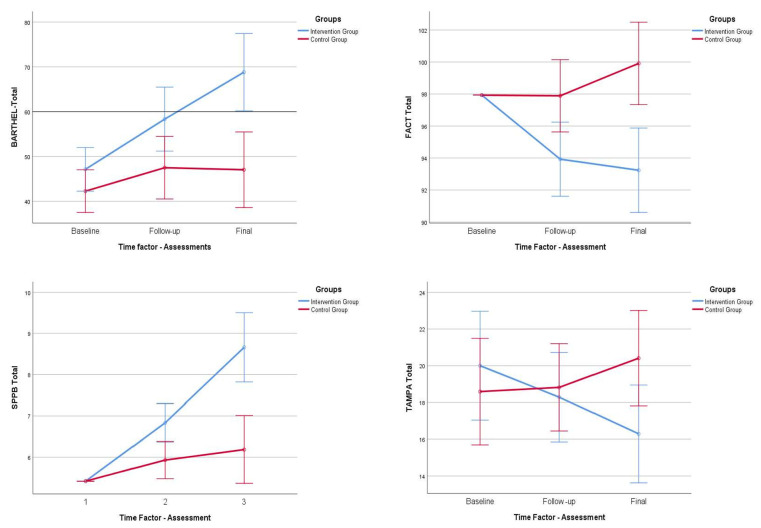
Within-group behavior and follow-up of score, Barthel Index, SPPB, TAMPA, and FACT-An. Comparisons between both groups. IG, Intervention group; CG, control group; FACT-An, Functional Evaluation of Cancer Therapy-Anemia; SPPB, short physical performance battery.

**Figure 3 ijerph-20-04938-f003:**
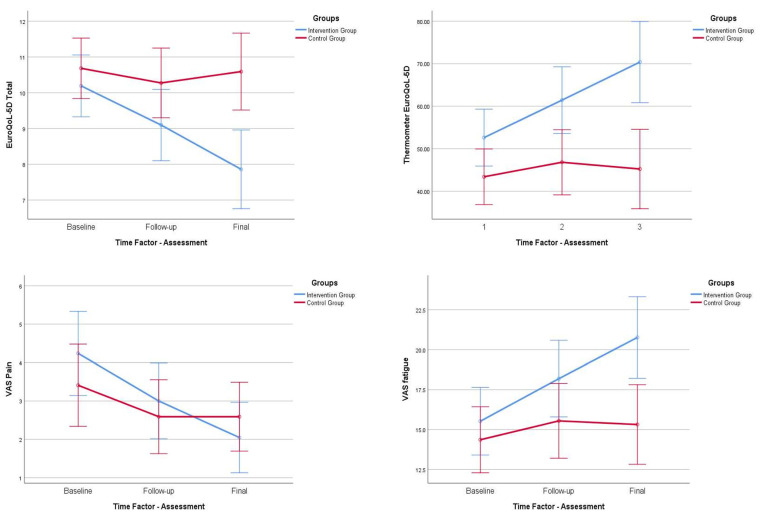
Within-group behavior and follow-up of score, EuroQoL 5D, VAS pain, and VAS fatigue scale. Comparisons between both groups. IG, intervention group; CG, control group; QoL, quality of life; VAS, visual analogic scale.

**Table 1 ijerph-20-04938-t001:** Sociodemographic variables of the sample at baseline evaluation.

Socio-Demographic Variables	Intervention Groups (Mean ± Standard Deviation)			
	Intervention Group (n = 24)	Control Group (n = 24)	Group Differences (*p*-Value)	
Men	n = 14	n = 16		
Women	n = 10	n = 8		
Age (years)	61.13 ± 13.33	65.79 ± 11.40	−4.667	(0.199)
Weight (kg)	65.70 ± 13.41	65.61 ± 13.45	−1.0600	(0.682)
Height (cm)	169.92 ± 7.76	168.83 ± 8.61	0.01220	(0.418)
BMI (kg/m^2^)	22.58 ± 3.23	22.75 ± 3.56	−0.75189	(0.459)
Number treatment lines	3.17 ± 1.49	3.46 ± 1.71	−0.292	(0.533)
Number of cohabitants	2 *	2 *	−0.417	(0.236)
Charlson comorbidity index	9.50 ± 4.12	9.96 ± 3.18	−0.458	(0.669)
Estimated survival 10 years	16.16 ± 32.56	9.96 ± 3.18	9.725	(0.213)

* median.

**Table 2 ijerph-20-04938-t002:** Outcome parameters at baseline evaluation and differences between groups.

Variables	Intervention Groups (Mean ± Standard Deviation)			
	Intervention Group (n = 24)	Control Group (n = 24)	Group Differences (*p*-Value)	
Barthel	47.08 ± 10.47	40.63 ± 12.79	6.458	(0.061)
FACT-An total	94.96 ± 11.91	102.75 ± 13.00	−7.792	(0.036) *
FACT physical state	15.42 ± 4.13	16.00 ± 3.98	−0.583	(0.621)
FACT: social environment	17.00 ± 4.89	16.71 ± 4.00	0.292	(0.822)
FACT: emotional state	12.79 ± 3.32	14.71 ± 3.39	−1.917	(0.054)
FACT: personal functioning	12.79 ± 3.32	10.88 ± 4.48	1.917	(0.125)
FACT: other concerns	37.79 ± 8.62	45.21 ± 8.66	−7.417	(0.005) *
EuroQoL-5D Total	10.33 ± 1.78	10.88 ± 2.19	−0.542	(0.353)
EuroQoL-5D mobility	2.04 ± 0.55	2.33 ± 0.48	−0.292	(0.057)
EuroQoL-5D personal care	2.21 ± 0.50	2.38 ± 0.47	−0.167	(0.294)
EuroQoL-5D daily activities	2.33 ± 0.48	2.54 ± 0.50	−0.208	(0.152)
EuroQoL-5D pain	1.96 ± 0.69	1.71 ± 0.62	0.250	(0.195)
EuroQoL-5D anxiety	1.79 ± 0.77	1.92 ± 0.77	−0.125	(0.580)
Thermometer EuroQoL-5D	50.83 ± 16.46	41.66 ± 15.01	9.16	(0.050)
VAS pain	4.29 ± 2.21	3.58 ± 2.78	0.708	(0.334)
VAS fatigue	15.17 ± 5.30	13.79 ± 4.66	1.375	(0.345)
SPPB total	6.21 ± 2.99	4.42 ± 3.03	1.792	(0.045) *
SPPB balance	2.58 ± 1.28	1.92 ± 1.17	0.667	(0.067)
SPPB gait	1.75 ± 0.94	1.17 ± 0.91	0.583	(0.035)
SPPB speed get up	1.88 ± 0.94	1.33 ± 1.12	0.542	(0.078)
TAMPA total	20.25 ± 7.07	18.71 ± 5.90	1.542	(0.417)
TAMPA avoidance	11.71 ± 4.57	10.88 ± 3.26	0.833	(0.471)
TAMPA damage	8.54 ± 3.32	7.83 ± 3.18	0.455	(0.708)

* statistically significant *p* < 0.05. Indicates significant differences between groups. m, mean; SD, standard deviation; IG, intervention group; CG, control group; FACT-An, Functional Evaluation of Cancer Therapy-Anemia; QoL, quality of life; SPPB, short physical performance battery; VAS, visual analogic scale.

**Table 3 ijerph-20-04938-t003:** Analytical statistics of the different variables under study. Baseline, follow-up, and final assessment in both groups (experimental and control) (two-way repeated measures ANOVA).

Variable	Intervention Group (IG)						Control Group (CG)					
	Baseline		Follow-Up (15 Days)		Final (30 Days)		Baseline		Follow-Up (15 Days)		Final (30 Days)	
BARTHEL	m	SD	MD	*p*-Value	MD	*p*-Value	m	SD	MD	*p*-Value	MD	*p*-Value
	47.14	±10.19	11.19	*p* < 0.001 *	21.66	*p* < 0.001 *	42.27	±11.82	5.22	*p* = 0.015 *	4.77	*p* = 0.180
			95% CI (Lower-Upper)		95% CI (Lower-Upper)				95% CI (Lower-Upper)		95% CI (Lower-Upper)	
			6.674, 15.707		15.363, 27.970				0.814, 9.640		−1.386, 10.931	
FACT	m	SD	MD	*p*-Value	MD	*p*-Value	m	SD	MD	*p*-Value	MD	*p*-Value
	94.95	±11.89	−4.00	*p* = 0.004 *	−4.69	*p* = 0.003 *	100.77	±11.01	−0.04	*p* = 1.000	1.98	0.383
			95% CI (Lower-Upper)		95% CI (Lower-Upper)				95% CI (Lower-Upper)		95% CI (Lower-Upper)	
			−6.865, −1.141		−7.959, −1.433				−2.837, 2.752		−1.203, 5.168	
FACT Physical condition	m	SD	MD	*p*-Value	MD	*p*-Value	m	SD	MD	*p*-Value	MD	*p*-Value
	15.29	±4.23	−2.14	*p* < 0.001 *	−3.85	*p* < 0.001 *	15.77	±4.02	−0.63	*p* = 0.195	-	*p* = 1.000
			95% CI (Lower-Upper)		95% CI (Lower-Upper)				95% CI (Lower-Upper)		95% CI (Lower-Upper)	
			−3.001, −1.285		−5.032, −2.683				−1.474, 0.202		−1.147, 1.147	
FACT social environment	m	SD	MD	*p*-Value	MD	*p*-Value	m	SD	MD	*p*-Value	MD	*p*-Value
	16.95	±5.16	0.90	*p* < 0.001 *	1.95	*p* < 0.001 *	16.45	±4.09	0.00	*p* = 1.000	−0.04	*p* = 1.000
			95% CI (Lower-Upper)		95% CI (Lower-Upper)				95% CI (Lower-Upper)		95% CI (Lower-Upper)	
			0.368, 1.442		1.182, 2.722				−0.524, 0.524		−0.798, 0.707	
FACT emotional state	m	SD	MD	*p*-Value	MD	*p*-Value	m	SD	MD	*p*-Value	MD	*p*-Value
	12.57	±3.32	−0.57	*p* = 0.120	−1.71	*p* = 0.001 *	14.23	±3.31	0.40	*p* = 0.383	1.18	*p* = 0.024
			95% CI (Lower-Upper)		95% CI (Lower-Upper)				95% CI (Lower-Upper)		95% CI (Lower-Upper)	
			−1.244, 0.101		−2.796, −0.633				−0.248, 1.066		0.125, 2.238	
FACT personal functioning	m	SD	MD	*p*-Value	MD	*p*-Value	m	SD	MD	*p*-Value	MD	*p*-Value
	13.29	±3.79	1.28	*p* < 0.001 *	2.85	*p* < 0.001 *	1.64	±4.48	0.72	*p* = 0.013 *	0.22	*p* = 1.000
			95% CI (Lower-Upper)		95% CI (Lower-Upper)				95% CI (Lower-Upper)		95% CI (Lower-Upper)	
			0.668, 1.903		1.734, 3.981				0.668, 1.903		1.734, 3.981	
FACT other concerns	m	SD	MD	*p*-Value	MD	*p*-Value	m	SD	MD	*p*-Value	MD	*p*-Value
	37.81	±7.93	−2.60	*p* < 0.001 *	−4.25	*p* < 0.001 *	44.50	±8.44	−1.65	*p* = 0.002 *	−0.80	*p* = 0.867
			95% CI (Lower-Upper)		95% CI (Lower-Upper)				95% CI (Lower-Upper)		95% CI (Lower-Upper)	
			−3.740, −1.470		−6.169, −2.337				−2.757, −0.543		−2.672, 1.065	
EuroQoL	m	SD	MD	*p*-Value	MD	*p*-Value	m	SD	MD	*p*-Value	MD	*p*-Value
	10.19	±1.72	−1.09	*p* < 0.001 *	−2.33	*p* < 0.001 *	10.68	±1.721	−0.40	*p* = 0.169	−0.09	*p* = 1.000
			95% CI (Lower-Upper)		95% CI (Lower-Upper)				95% CI (Lower-Upper)		95% CI (Lower-Upper)	
			−0.627, −0.563		−3.027, −1.640				−0.929, 0.111		−0.769, 0.587	
EuroQoL Thermometer	m	SD	MD	*p*-Value	MD	*p*-Value	m	SD	MD	*p*-Value	MD	*p*-Value
	52.61	±15.93	8.810	*p* < 0.001 *	17.76	*p* < 0.001 *	43.409	±14.42	3.40	*p* = 0.255	1.81	*p* = 1.000
			95% CI (Lower-Upper)		95% CI (Lower-Upper)				95% CI (Lower-Upper)		95% CI (Lower-Upper)	
			3.875, 13.744		11.106, 24.418				−1.412, 8.230		−4.684, 8.321	
VAS pain	m	SD	MD	*p*-Value	MD	*p*-Value	m	SD	MD	*p*-Value	MD	*p*-Value
	4.24	±2.18	−1.23	*p* < 0.001 *	−2.19	*p* < 0.001 *	3.41	±2.737	−0.81	*p* = 0.021 *	−0.82	*p* = 0.112
			95% CI (Lower-Upper)		95% CI (Lower-Upper)				95% CI (Lower-Upper)		95% CI (Lower-Upper)	
			−1.975, −0.501		−3.162, −1.219				−1.538, −0.098		−0.649, 0.649	
VAS scale	m	SD	MD	*p*-Value	MD	*p*-Value	m	SD	MD	*p*-Value	MD	*p*-Value
	15.52	±5.18	2.66	*p* < 0.001 *	5.23	*p* < 0.001 *	14.36	±4.414	1.18	*p* = 0.032 *	0.95	*p* = 0.384
			95% CI (Lower-Upper)		95% CI (Lower-Upper)				95% CI (Lower-Upper)		95% CI (Lower-Upper)	
			1.538, 3.795		3.668, 6.808				0.079, 2.284		−0.580, 2.489	
SPPB	m	SD	MD	*p*-Value	MD	*p*-Value	m	SD	MD	*p*-Value	MD	*p*-Value
	6.29	±3.05	1.41	*p* < 0.001 *	3.24	*p* < 0.001 *	4.59	±3.112	0.51	*p* = 0.082	0.76	*p* = 0.199
			95% CI (Lower-Upper)		95% CI (Lower-Upper)				95% CI (Lower-Upper)		95% CI (Lower-Upper)	
			0.841, 1.989		2.201, 4.285				−0.047, 1.073		−0.249, 1.785	
SPPB balance	m	SD	MD	*p*-Value	MD	*p*-Value	m	SD	MD	*p*-Value	MD	*p*-Value
	2.67	±1.31	0.23	*p* = 0.172	0.52	*p* = 0.014	2.00	±1.19	0.04	*p* = 1.000	0.04	*p* = 1.000
			95% CI (Lower-Upper)		95% CI (Lower-Upper)				95% CI (Lower-Upper)		95% CI (Lower-Upper)	
			−0.066, 0.542		0.088, 0.959				−0.251, 0.342		−0.380, 0.471	
SPPB gait	m	SD	MD	*p*-Value	MD	*p*-Value	m	SD	MD	*p*-Value	MD	*p*-Value
	1.71	±0.95	0.58	*p* < 0.001 *	1.38	*p* < 0.001 *	1.23	±0.922	0.30	*p* = 0.022 *	0.40	*p* = 0.036 *
			95% CI (Lower-Upper)		95% CI (Lower-Upper)				95% CI (Lower-Upper)		95% CI (Lower-Upper)	
			0.312, 0.861		0.944, 1.732				0.036, 0.572		0.020, 0.789	
SPPB speed get up	m	SD	MD	*p*-Value	MD	*p*-Value	m	SD	MD	*p*-Value	MD	*p*-Value
	1.90	±0.94	0.52	*p* < 0.001 *	1.28	*p* < 0.001 *	1.36	±1.17	0.18	*p* = 0.299	0.40	*p* = 0.088
			95% CI (Lower-Upper)		95% CI (Lower-Upper)				95% CI (Lower-Upper)		95% CI (Lower-Upper)	
			0.248, 0.800		0.823, 1.174				−0.088, 0.451		−0.043, 0.861	
TAMPA	m	SD	MD	*p*-Value	MD	*p*-Value	m	SD	MD	*p*-Value	MD	*p*-Value
	20.00	±7.34	−1.71	*p* = 0.005 *	−3.714	*p* = 0.001 *	18.59	±6.10	0.22	*p* = 1.000	1.81	*p* = 0.176
			95% CI (Lower-Upper)		95% CI (Lower-Upper)				95% CI (Lower-Upper)		95% CI (Lower-Upper)	
			−2.97, −0.45		−6.10, −1.32				−1.00, 1.46		−0.51, 4.15	
TAMPA avoidance	m	SD	MD	*p*-Value	MD	*p*-Value	m	SD	MD	*p*-Value	MD	*p*-Value
	11.48	±4.57	−0.81	*p* = 0.166	−1.714	*p* = 0.046 *	10.95	±3.40	0.77	0.183	1.50	0.087
			95% CI (Lower-Upper)		95% CI (Lower-Upper)				95% CI (Lower-Upper)		95% CI (Lower-Upper)	
			−1.83, 0.21		−3.40, −0.02				−0.22, 1.77		−0.15, 3.15	
TAMPA damage	m	SD	MD	*p*-Value	MD	*p*-Value	m	SD	MD	*p*-Value	MD	*p*-Value
	8.52	±3.55	−0.90	*p* = 0.006 *	−1.952	*p* < 0.001 *	7.64	±3.12	−0.13	*p* = 1.000	0.22	*p* = 1.000
			95% CI (Lower-Upper)		95% CI (Lower-Upper)				95% CI (Lower-Upper)		95% CI (Lower-Upper)	
			−1.58, −0.22		−3.05, −0.84				−0.80, 0.53		−0.45, 1.18	

* statistically significant.

## Data Availability

The data presented in this study are available on reasonable request from the corresponding author. The data are not publicly available due to the applicable data protection law.

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
