# Peer review of "Multimodal Physical Exercise and Functional Rehabilitation Program in Oncological Patients with Cancer-Related Fatigue—A Randomized Clinical Trial"

_ijerph, 2023, doi:10.3390/ijerph20064938_

Round 1

Reviewer 1 Report (Previous Reviewer 1)

Thank you for revising your manuscript.

#1: Although 2 rounds of review, the program adherence is still unclear. Have all participants in the experimental group performed the exercise program every day for one month?

#2: Thank you for adding the clinical backgrounds of the participants. Was there any significant difference in clinical backgrounds between the two groups?

#3: In the discussion section, paragraph 8, from line 496, the authors indicated a direct relationship between functional capacity and kinesiophobia, but there was no statistical analysis data.

Author Response

REVIEWER 1:

Thank you for revising your manuscript.

Response: Thank you very much for your comments and suggestions. Based on them, we believe that the manuscript has improved in both quality and clarity of exposition. Below is the response to your more specific comments. Thank you very much.

 #1: Although 2 rounds of review, the program adherence is still unclear. Have all participants in the experimental group performed the exercise program every day for one month?

Response: Thank you very much for your comments. We explain the adherence to the intervention in a new paragraph at the end of the discussion section. Hopefully this will make it more correct. Thank you very much.

 #2: Thank you for adding the clinical backgrounds of the participants. Was there any significant difference in clinical backgrounds between the two groups?

Response: Thank you very much for your comments. No significant difference was detected between the clinical histories of the two groups. There is a significant similarity in the clinical history of both the control and experimental groups.

 #3: In the discussion section, paragraph 8, from line 496, the authors indicated a direct relationship between functional capacity and kinesiophobia, but there was no statistical analysis data.

Response: Thank you very much for your comment. We have changed the wording as "direct relationship" can be misleading. We refer to improvements in both the functional capacity and kinesiophobia of patients. We have changed the wording in the text so that it does not create controversy. Thank you very much for your comment.

Reviewer 2 Report (Previous Reviewer 2)

I have no more questions for the revised manuscript.

Author Response

REVIEWER 2:

I have no more questions for the revised manuscript.

Response: Thank you very much for your comments. Your suggestions are greatly appreciated and have considerably improved the submitted manuscript. It has been a pleasure.

This manuscript is a resubmission of an earlier submission. The following is a list of the peer review reports and author responses from that submission.

Round 1

Reviewer 1 Report

This study examines whether physical exercise and functional rehabilitation improve cancer-related fatigue and asthenia at home after discharge from the hospital. It measures QOL and is considered an important study. It is very easy to understand and highly reproducible, but some corrections are desirable.

#1: For enrolled patients, did authors collect the clinical backgrounds of participants such as the type of cancer, treatment history, stage, and reason for admission to the hospital? It would be easier to understand if Table 1 included social backgrounds such as the male-female ratio and educational background, which are only mentioned in the text.

#2: Did the intervention program run daily for a month? How well did participants in the intervention group adhere to the program? It would be easier to understand if the duration of the intervention program was also stated in the abstract.

#3: Could you explain the reason why the authors did not use the EQ-5D-5L tariff?

#4: Related to #1, the background of the participants is unclear, making it difficult to consider the content of the discussion.

Author Response

POINT-BY-POINT RESPONSE TO REVIEWER 1

REVIEWER 1:

This study examines whether physical exercise and functional rehabilitation improve cancer-related fatigue and asthenia at home after discharge from the hospital. It measures QOL and is considered an important study. It is very easy to understand and highly reproducible, but some corrections are desirable.

 #1: For enrolled patients, did authors collect the clinical backgrounds of participants such as the type of cancer, treatment history, stage, and reason for admission to the hospital? It would be easier to understand if Table 1 included social backgrounds such as the male-female ratio and educational background, which are only mentioned in the text.

Response: Thank you very much for the suggestion. We will make the changes as you have told us. Gender-related data are included in table 1.

#2: Did the intervention program run daily for a month? How well did participants in the intervention group adhere to the program? It would be easier to understand if the duration of the intervention program was also stated in the abstract.

Response: Thank you very much for the suggestion. We will make the changes as you have told us. We add the duration of the intervention in the abstract.

#3: Could you explain the reason why the authors did not use the EQ-5D-5L tariff?

Response: We decided not to include the results of this variable in the study because we did not get good feedback from the patients at the time of the evaluations. We think that it may not be the most reliable measuring instrument.

#4: Related to #1, the background of the participants is unclear, making it difficult to consider the content of the discussion.

Response: Thank you very much for the input. We will make the changes as you have told us.

Reviewer 2 Report

Thanks for the invitation to review the manuscript 'Multimodal physical exercise and functional rehabilitation program in oncological patients with cancer-related fatigue. A randomized clinical trial'. This study evaluated the effectiveness of a multimodal rehabilitation program on asthenia, pain, functional capacity and QoL in individuals with oncology conditions. The study revealed a beneficial effect of the program.

The study fits into the scope of the special issue. In general, the manuscript is easy to follow. The study has a high impact on oncology rehabilitation. The major concern for me is that the data analysis is not conducted in accordance with that described in the methodology. Some information is missing in the method section. Please find below my specific comments and suggestions.

Introduction

1.         Line 83 – 84 There is an existing study adopting the multimodal rehabilitation program to manage cancer-related fatigue. Eg. Kröz et.al 2017. The authors should try to describe these studies in the introduction

Methods

2.         The authors have not mentioned whether the data from drop-out participants were included in the analysis and how they managed the missing data

Results

3.         The number of participants recruited is fewer than that in the original sample size estimation. The authors should provide a brief explanation.

4.         In the sample size estimation, the expected BI changes of 7.5 points did not match with that described in the previously published protocol paper (17 points)

5.         For each group, report the reason of losses and exclusions after randomization.

6.         Report the dates defining the periods of recruitment and follow-up

7.         The authors mentioned that a two-way repeated measures ANOVA is used to analyze the data. However, it seems that only the within-group differences were reported. Information regarding the between-group and interaction effect has not been reported.

8.         For figure 2 and 3, please label the 4 subplots and quote accordingly in the text

9.         The Tables and figures should be placed in the section where they were quoted in the text. Not under the section ‘Figures, Tables and Schemes’.

Discussion

10.     Please comment on the generalizability of the results

11.     Line 466: the effect of such program on cancer pain has been explored in this study

Kröz, M.et al (2017). Impact of a combined multimodal-aerobic and multimodal intervention compared to standard aerobic treatment in breast cancer survivors with chronic cancer-related fatigue-results of a three-armed pragmatic trial in a comprehensive cohort design. BMC cancer, 17(1), 1-13.

Author Response

POINT-BY-POINT RESPONSE TO REVIEWER 2

REVIEWER 2:

Thanks for the invitation to review the manuscript 'Multimodal physical exercise and functional rehabilitation program in oncological patients with cancer-related fatigue. A randomized clinical trial'. This study evaluated the effectiveness of a multimodal rehabilitation program on asthenia, pain, functional capacity and QoL in individuals with oncology conditions. The study revealed a beneficial effect of the program.

The study fits into the scope of the special issue. In general, the manuscript is easy to follow. The study has a high impact on oncology rehabilitation. The major concern for me is that the data analysis is not conducted in accordance with that described in the methodology. Some information is missing in the method section. Please find below my specific comments and suggestions.

Introduction

  1. Line 83 – 84 There is an existing study adopting the multimodal rehabilitation program to manage cancer-related fatigue. Eg. Kröz et.al 2017. The authors should try to describe these studies in the introduction.

Response: Thank you very much for the suggestion. We will make the changes as you have told us. We add the contents of this important study in the introduction and in the "bibliography" section.

Methods

  1. The authors have not mentioned whether the data from drop-out participants were included in the analysis and how they managed the missing data

Response: Thank you very much for the input/suggestion. Data from participants who were excluded or dropped out of the study were not considered for analysis. We add this information in the section "methods".

Results

  1. The number of participants recruited is fewer than that in the original sample size estimation. The authors should provide a brief explanation.

Response: Thank you very much for the input/suggestion. The number of patients recruited is 48, distributed in 24 patients in each group (experimental and control), as calculated in the sample size estimation. During the development of the intervention, 5 losses occurred, all of them with a common reason, the patient's exitus, as shown in figure 1.

  1. In the sample size estimation, the expected BI changes of 7.5 points did not match with that described in the previously published protocol paper (17 points)

Response: Thank you very much for the suggestion. We apologise for this error in the published protocol, it is not our mistake, it is an error in the final layout of the article, in the coming weeks it will be corrected in the published protocol, it was an editing error of the journal, we have already contacted them a few weeks ago to fix it.

  1. For each group, report the reason of losses and exclusions after randomization.

Response: Thank you very much for the input/suggestion. These data are reflected in figure 1, corresponding to the flow chart of the study.

  1. Report the dates defining the periods of recruitment and follow-up.

Response: Thank you very much for the suggestion. These data are included in the "methods" section.

  1. The authors mentioned that a two-way repeated measures ANOVA is used to analyze the data. However, it seems that only the within-group differences were reported. Information regarding the between-group and interaction effect has not been reported.

Response: Thank you very much for the input/suggestion. We consider this analysis to be the most relevant for reporting the results of the study, as the space of the article is not large enough to address all the contents of the study.

  1. For figure 2 and 3, please label the 4 subplots and quote accordingly in the text.

Response: Thank you very much for the input/suggestion. We will make the changes as you have told us.

  1. The Tables and figures should be placed in the section where they were quoted in the text. Not under the section ‘Figures, Tables and Schemes’.

Response: Thank you very much for the input/suggestion. We have modified the distribution as you have told us, thank you very much.

Discussion

  1. Please comment on the generalizability of the results.

Response: Thank you very much for the input/suggestion. We will make the changes as you have told us.

  1. Line 466: the effect of such program on cancer pain has been explored in this study.

Kröz, M.et al (2017). Impact of a combined multimodal-aerobic and multimodal intervention compared to standard aerobic treatment in breast cancer survivors with chronic cancer-related fatigue-results of a three-armed pragmatic trial in a comprehensive cohort design. BMC cancer, 17(1), 1-13.

Response: Thank you very much for the input/suggestion. We will make the changes as you have told us.

Round 2

Reviewer 1 Report

Thank you for your revision of your manuscript.

In this revised manuscript, there are still insufficient clinical backgrounds (such as type of cancer, stage, time since diagnosis, treatment history of cancer, etc.) of the study participants to do a clinical appraisal of the content of the paper. Also, adherence to the intervention program was unclear from the results.

Reviewer 2 Report

1. For the missing data, the authors clarified that data from participants who dropped out of the study were not included in the data analysis. This approach may have violated the 'intention-to-treat' principle. The intention-to-treat principle requires that all study participants be included in the analyses, regardless of whether the participant received any exposure to the assigned study treatment or complied with the treatment. As the authors mention that the analyses were conducted on an intention-to-treat basis (Line 252). There is a contradiction in the descriptions.

 2. Regarding statistical results, as the authors decided to use repeated measured ANOVA to analyse the data. It is essential for the authors to report the corresponding statistical results. Traditionally, the reporting should include the fixed effect of group and time and the interaction effect. Please provide a brief explanation if the interaction effect was not included in the original statistical model. Reporting the within-group differences only could be miss leading if the interaction effect is not significant.